# Design and Simple Assembly of Gold Nanostar Bioconjugates for Surface-Enhanced Raman Spectroscopy Immunoassays

**DOI:** 10.3390/nano9111561

**Published:** 2019-11-04

**Authors:** Maria João Oliveira, Miguel P. de Almeida, Daniela Nunes, Elvira Fortunato, Rodrigo Martins, Eulália Pereira, Hugh J. Byrne, Hugo Águas, Ricardo Franco

**Affiliations:** 1UCIBIO, REQUIMTE, Departamento de Química, Faculdade de Ciências e Tecnologia, Universidade NOVA de Lisboa, 2829-516 Caparica, Portugal; mj.oliveira@campus.fct.unl.pt; 2CENIMAT-I3N, Departamento de Ciência dos Materiais, Faculdade de Ciências e Tecnologia, FCT, Universidade Nova de Lisboa, 2829-516 Caparica, Portugal; daniela.gomes@fct.unl.pt (D.N.); emf@fct.unl.pt (E.F.); rfpm@fct.unl.pt (R.M.); 3REQUIMTE/LAQV, Departamento de Química e Bioquímica, Faculdade de Ciências da Universidade do Porto, 4169-007 Porto, Portugal; mpda@fc.up.pt (M.P.d.A.); eulalia.pereira@fc.up.pt (E.P.); 4FOCAS Research Institute, Technological University Dublin, Kevin Street, Dublin 8, Ireland; hugh.byrne@tudublin.ie

**Keywords:** surface enhanced Raman spectroscopy Surface, Enhanced Raman Spectroscopy, gold nanostars, gold nanoparticles, immunoconjugates, immunoassay, horseradish peroxidase, Raman reporter, agarose gel electrophoresis, dynamic light scattering

## Abstract

Immunoassays using Surface-Enhanced Raman Spectroscopy are especially interesting on account not only of their increased sensitivity, but also due to its easy translation to point-of-care formats. The bases for these assays are bioconjugates of polyclonal antibodies and anisotropic gold nanoparticles functionalized with a Raman reporter. These bioconjugates, once loaded with the antigen analyte, can react on a sandwich format with the same antibodies immobilized on a surface. This surface can then be used for detection, on a microfluidics or immunochromatographic platform. Here, we have assembled bioconjugates of gold nanostars functionalized with 4-mercaptobenzoic acid, and anti-horseradish peroxidase antibodies. The assembly was by simple incubation, and agarose gel electrophoresis determined a high gold nanostar to antibody binding constant. The functionality of the bioconjugates is easy to determine since the respective antigen presents peroxidase enzymatic activity. Furthermore, the chosen antibody is a generic immunoglobulin G (IgG) antibody, opening the application of these principles to other antibody-antigen systems. Surface-Enhanced Raman Spectroscopy analysis of these bioconjugates indicated antigen detection down to 50 µU of peroxidase activity. All steps of conjugation were fully characterized by ultraviolet-visible spectroscopy, dynamic light scattering, ζ-Potential, scanning electron microscopy, and agarose gel electrophoresis. Based on the latter technique, a proof-of-concept was established for the proposed immunoassay.

## 1. Introduction

Raman spectroscopy has been intensively studied as a non-destructive and sensitive technique for in situ biological applications, due to the low background signal of water [1,2,3]. However, it is intrinsically a weak phenomenon. Due to interactions from the analyte adsorbed on noble metal nanoparticles (NPs) that are excited near their surface plasmon resonance, Surface Enhanced Raman Scattering (SERS) has the potential to be an efficient method of bio-detection [4,5]. In fact, SERS increases the sensitivity of Raman scattering dramatically, which enables analyte identification without involving labels, complex chemistry, or time-consuming steps, and rivals the sensitivity of fluorescence-based detection techniques [4]. On the other hand, the use of reporter-based methods, i.e., SERS tags overcomes the challenging interpretation of intrinsic SERS spectra of biological specimens. For protein antigen detection applications, the most useful strategy for improved sensitivity is a Raman reporter molecule coupled to a silver or gold NP and conjugated with a specific antibody [5,6]. The SERS spectral signature from the Raman reporter, thus, reflects the presence of the antigen, with its area or intensity being proportional to antigen concentration. The excellent enhancement factors presented by Raman reporters provide an easier way to perform quantitative assays and achieve high sensitivity and lower detection limits. The distinctive and narrow Raman lines from such reporters minimize the spectroscopic overlap, which allows multiple label testing [7,8]. Nevertheless, poor reproducibility and uniformity under complicated experimental conditions still need to be addressed [4]. The lack of standardization methods for characterization of SERS-tag reagents hinders a more correct analysis of an assay’s analytical performance [9,10]. For instance, the enhancement factor is frequently used as a property of SERS-tags, but concentration, size heterogeneity, and an external intensity standard for comparison may present limitations for using this approach.

The most common analytical tool for biochemical studies and clinical diagnosis is the immunoassay. One interesting format relies on the formation of a sandwich complex between polyclonal antibodies immobilized on a surface and the same polyclonal antibodies conjugated with a Raman-labelled SERS-active gold NP [5,6,11]. For a positive test result, the antigen binds to antibodies on both surfaces, which makes the surface SERS-active after washing. Thus, this scheme is particularly suitable for a microfluidic type of platform [6].

The NP surface is usually modified to provide a bio-friendly environment (such as alkanethiols for gold surfaces). Nevertheless, the mechanism of interaction between proteins and noble metal NPs needs to be carefully assessed for each specific NP-antibody pair, for the particular intended use in nanomedicine [12]. Conformational changes of the proteins resulting from interactions with metal NPs can affect physicochemical properties of each counterpart, which leads to a loss of reactivity and impaired applicability of the bioconjugates in detection methods. A broad variety of techniques such as ultraviolet-visible – near infrared (UV-Vis-NIR) absorption [13], fluorescence [14], differential centrifugal sedimentation (DCS) [15], dynamic light scattering (DLS) [16], and circular dichroism spectroscopies [17,18,19] or even SERS [20] can elucidate the mechanism for these interactions. Several studies have demonstrated enhanced enzymatic activities for enzymes conjugated with NPs [12,17,21]. For example, laccase electrostatically conjugated with spherical gold NPs proved to be nine times more active than the free enzyme [15]. Other examples include the observation that lysozyme did not experience perturbations to its structure and function, but silver NPs underwent aggregation with increasing protein amounts [18]. These reports focused on the final fate of the bioconjugate structure and function and its applicability. Nevertheless, they highlight the importance of a deep understanding of physicochemical properties of the NP-biomolecule system for successful and reproducible design of bioconjugates.

Herein, we present the design, simple assembly process, and testing of bioconjugates based on gold nanostars (AuNSs) and antibodies, for immuno-detection SERS applications. Antibodies are conjugated with functionalized AuNSs by a simple incubation method, which relies on establishing non-covalent interactions with the bifunctional ligand that is also a Raman reporter. This type of conjugate can be expanded for multiplexing detection using several different Raman reporters [11]. 

Gold nanostars are nanoparticles presenting sharp branches emanating from a core. The presence of several tips provides “hot spots”, which makes them an ideal candidate for SERS applications [22,23,24,25]. Moreover, their plasmonic properties can even be controlled by modulating the morphology of AuNSs arms and, consequently, enhancing the SERS signal [26].

In this work, we demonstrate a simple method for optimizing and testing bioconjugates for SERS-based immunoassays. Our bioconjugates consisted of AuNSs with a monolayer of the thiol and Raman reporter 4-mercaptobenzoic acid (MBA). In this way, MBA confers a distinct spectral signature and, simultaneously, a stabilizing coating to the nanoparticles and a surface for conjugation with a molecular recognition element, the antibody. Electrostatic interactions were promoted by the carboxylic group of MBA and the amine group of the antibody. The polyclonal antibody chosen was anti-Horseradish peroxidase (Anti-HRP) in order to easily access the viability of the bioconjugate through an enzymatic assay with horseradish peroxidase, which also functions as the antigen analyte [27]. Lastly, the SERS signal of the bioconjugates was analyzed to determine the minimum concentration detected. Agarose gel electrophoresis (AGE) was used to easily optimize the amount of antibody to load onto the AuNSs. Moreover, AGE allowed us to establish a proof-of-concept for the proposed sandwich immunoassay. The bioconjugates formed in aqueous solution were characterized by ultraviolet-visible spectroscopy (UV-Vis), DLS, and ζ-potential measurements.

## 2. Materials and Methods

### 2.1. Reagents and Materials

The following reagents were used for the synthesis and functionalization of gold nanostars: Gold(III) chloride solution 30% wt. Au in dilute HCl (99.99%, HAuCl_4_, Sigma-Aldrich, St. Louis, MO, USA), sodium citrate tribasic dihydrate (99.0%, C_6_H_5_Na_3_O_7_ 2H_2_O, Sigma-Aldrich St. Louis, MO, USA), hydrochloric acid (HCl), silver nitrate (99.9999%, AgNO_3_, Sigma-Aldrich, St. Louis, MO, USA), L-ascorbic acid (99.9%, C_6_H_8_O_6_, Fluka, Buchs, Switzerland), nitric acid (65%, PanReac AppliChem, Gatersleben, Germany), and hydrochloric acid (37%, Fisher Chemical, Loughborough, UK). Tris(hydroxymethyl)-aminomethane, ethylenediamine tetra-acetic acid (EDTA), acetic acid (CH_3_COOH), and the Raman reporter of 4-mercaptobenzoic acid were purchased from Sigma-Aldrich and used without further purification. Ultrapure water (18 MΩ.cm) was used for preparing all solutions unless stated otherwise. The anti-HRP antibody for bioconjugation of AuNSs and its antigen, HRP (horseradish peroxidase), were from Antibodies-Online—Germany and Sigma-Aldrich, respectively. Bovine serum albumin was also purchased from Sigma-Aldrich. Protein determination was by the bicinchoninic acid method (based on Smith et al. [28]) using a kit from Sigma-Aldrich—St. Louis, MO, USA. UltraPure agarose for agarose gels was from Invitrogen—Thermo Fisher Scientific, Waltham, MA, USA. Unless otherwise stated, all other chemicals and reagents were of the highest purity available.

### 2.2. Gold Nanoparticles Synthesis and Functionalization

All glassware used for the synthesis of NPs was previously immersed in freshly prepared *aqua regia*, a 1:3 mixture of nitric acid (HNO_3_, Panreac AppliChem, Gatersleben, Germany), and hydrochloric acid (HCl, Fisher Chemical). Afterward, the glassware was vigorously washed with ultrapure water (18.2 MΩ.cm at 25 °C) prior to use. As proposed by Ojea-Jiménez et al. [29] as a variation of the conventional Turkevich synthesis protocol, 2 mL of a 343 mM trisodium citrate (Sigma-Aldrich, St. Louis, MO, USA) solution were added to 98 mL of Milli-Q water in a round-bottom flask under heating and vigorous stirring (≈700 rpm) using a magnetic stirrer, while being kept away from sunlight. After boiling, 69.2 µL of a 1.445 M HAuCl_4_ solution (30 wt. % Au (III) chloride in dilute HCl, Sigma-Aldrich, St. Louis, MO, USA) were added to start the gold reduction reaction. After 5 min, the heating was stopped, and the suspension was cooled down to room temperature. Lastly, the suspension was transferred to a glass vial covered with aluminum foil and stored in the dark at 4 °C until further use.

After gold nanoparticles (AuNPs) synthesis and characterization, this suspension was used as seed for AuNSs synthesis. Surfactant-free AuNSs were synthesized using the seed-mediated growth method adapted from Yuan et al. [30]. A solution of 15.5 µL of HAuCl_4_ at 1.445 M was added to 7 mL of a 2 nM suspension of AuNPs (diameter of 12 nm). Then, 450 μL of a 100 mM ascorbic acid (Fluka Buchs, Switzerland) solution and 450 μL of a 4 mM silver nitrate (Sigma-Aldrich, St. Louis, MO, USA) solution were added simultaneously. The resulting suspension was gently stirred for 30 s before centrifugation for 15 min at 3000 g (Centurion Scientific K3 Series centrifuge) and resuspension in 10 mL of ultrapure water. The diameter and concentration of the spherical nanoparticles were determined according to the relation described by Haiss et al. [31]. The same parameters for star-shaped nanoparticles were determined by the method of Puig et al. [32]. 

Gold nanostars were functionalized with a Raman reporter, 4-mercaptobenzoic acid (MBA, Sigma-Aldrich, St. Louis, MO, USA). The molecule works simultaneously as a capping agent for the AuNSs, a Raman reporter, and a bio-friendly intermediate for antibody bioconjugation. A suitable volume of a 10 mM ethanolic solution of MBA (ethanol, Sigma-Aldrich, St. Louis, MO, USA) was added to the AuNSs suspension under vigorous stirring. The solution was allowed to react overnight at room temperature to ensure a complete formation of a self-assembled monolayer. The excess of MBA was removed through centrifugation at 2500 g for 10 min, which was followed by redispersion. Complete monolayer formation was assessed by Agarose Gel Electrophoresis. The AuNSs suspension UV-Vis spectrum was checked to confirm successful functionalization and to identify possible aggregation effects.

### 2.3. Bioconjugates Assembly

The AuNS–MBA–anti-HRP bioconjugates were prepared with the previously functionalized nanoparticles—at 0.2 nM—with appropriate amounts of antibody to obtain bioconjugates with the desired [AuNS]:[anti-HRP] molar ratio. Incubation was in 5 mM phosphate buffer at a pH of 7.2 for 90 min, in an orbital shaker at 250 rpm and 25 °C. Samples were then centrifuged for 10 min at 4 °C and 2500 g. The supernatant was discarded to remove excess protein and was resuspended in 5 mM phosphate buffer at a pH of 7.2 and were ready for use. In the case of AGE samples, pellets were resuspended only in 13.5 μL of 5 mM phosphate buffer at a pH of 7.2 and 1.5 μL of glycerol (Sigma-Aldrich, St. Louis, MO, USA). Bovine serum albumin (BSA, Sigma-Aldrich, St. Louis, MO, USA) was used to block non-specific interactions at the same molar ratio used for the anti-HRP. The AuNS–MBA–anti-HRP–BSA, is, hereafter, referred as “bioconjugates.” These bioconjugates were then used to incubate with HRP and/or anti-HRP to simulate the immunoassay. Incubation and wash steps were performed as above.

### 2.4. HRP Enzymatic Assay

In order to determine the viability of the produced anti-HRP-containing bioconjugates for antigen detection, its capacity to bind HRP antigens was tested via an HRP enzymatic assay [27]. Detection of peroxidase enzymatic activity was based in the Sigma-Aldrich protocol [33]. Measurements were carried out at 25 °C and pH = 5, Abs_405nm_, and light path = 1 cm. A 0.001 M potassium phosphate buffer (pH = 5.0) was prepared as a reference sample. The activity of the enzyme HRP (horseradish peroxidase, Sigma-Aldrich) was determined by monitoring the formation of the reaction product ABTS (2,2′-Azino-bis(3-ethylbenzthiazoline-6-sulfonic acid), Roche) that is oxidized in the presence of hydrogen peroxide (H_2_O_2_—Panreac AppliChem). This oxidation of ABTS is followed by measuring the absorbance at 405 nm by UV–Vis spectrophotometry. Enzyme activity was normalized to the free enzyme.

### 2.5. Ultraviolet-Visible Spectroscopy

All absorption spectra were performed in a UV-Vis spectrophotometer Cary 50 Bio (Varian^®^, San Francisco, CA, USA) using quartz cells with a 1-cm path length (Hellma^®^, Müllheim, Germany), with a wavelength range between 300 and 800 nm (scan rate of 600 nm/min), at room temperature. 

### 2.6. Light Scattering Measurements

Dynamic light scattering and ζ-potential measurements were performed in a SZ-100 Nanopartica series (Horiba, Japan). A 4 mW He–Ne laser (532 nm) was used with a fixed 90° scattering angle. All measurements were carried out at 25 °C, and the experiments started only after the sample reached thermal equilibrium (5 min). A volume of 60 µL was transferred to a cuvette for DLS (quartz cells with 3-mm path length from Hellma^®^, Müllheim, Germany) with a scattering angle equal to 90° or to a disposable ζ cell (Horiba, Japan) (ζ-Potential). In DLS, each sample was measured three times and each measurement consisted of 10 acquisitions. Cumulating statistics were used to measure the hydrodynamic diameter and polydispersity. In ζ-potential, each sample was measured three times and each measurement consisted of 100 acquisitions.

### 2.7. Scanning Electron Microscopy and X-ray Powder Diffraction

Scanning electron microscopy (SEM) observations of the AuNSs were carried out in a Carl Zeiss AURIGA Crossbeam (FIB-SEM) Workstation (Oberkochen, Germany) equipped for energy-dispersive spectroscopy (EDS) measurements. Samples were prepared by placing one drop of the nanoparticle’s solution on a silicon wafer and drying at room temperature.

The crystalline phases of the samples were verified using powder X-ray powder diffraction (XRD). 202 X’Pert PRO PANAlytical X-ray diffractometer (California, USA) was used to obtain X-ray diffraction patterns of the AuNSs. The 2θ values were taken from 15° to 80° using a Cu-Kα radiation (k = 1.54060 Å) with a step size of 0.033°. The Scherrer’s equation was used to measure the average crystallite size. Samples were prepared by placing one drop of the nanoparticle’s solution on a silicon wafer and drying at room temperature.

### 2.8. Agarose Gel Electrophoresis

Agarose gel electrophoresis was employed to determine the variations in charge and size, as previously reported for gold nanoparticles of different functionalities and, consequently, used as a tool to demonstrate the formation of the bioconjugates [12,15,34,35,36,37].

A horizontal agarose gel system was used in all experiments under a constant voltage of 150 V (E = 10 V/cm) in a mini-sub cell GT (Bio-Rad) with agarose from UltraPure™ Agarose, Invitrogen including 0.3% in Tris-acetate-EDTA (TAE) buffer 0.125×. Samples were incubated overnight in a 4 °C refrigerator, and then centrifuged at ~9500 g at 10 °C for 10 min, and the supernatant was discarded. Furthermore, 13.5 µL of potassium phosphate buffer (pH = 7.4, 5 mM) was used to resuspend the pellet. Lastly, 1.5 µL of glycerol was added to increase sample density and improve well deposition. Digital pictures of the gels were processed by eReuss software (see next section), which provided an accurate measurement of the red bands migration in agarose, and, thus, allowed the calculation of their electrophoretic mobility. Electrophoretic mobility (μ) is defined as the observed rate of migration of a component (ν) divided by the electric field strength (E) in a given medium. In the case of AGE, which is a solid support medium, only apparent values can be determined [34,38]. We represent our AGE mobilities as variations relative to the maximum mobility band (Δµ).

### 2.9. Adsorption Isotherm Fitting to AGE Data

As more antibodies are adsorbed at the functionalized AuNS surfaces, the electrophoretic mobility for the newly formed conjugate is reduced as its mass increases. Its surface loses some negative charge. This behavior is reflected in a reduced migration toward the positive electrode. Eventually, the mobility reaches a plateau corresponding to saturation of the AuNS-conjugate surface with the antibody. Using eReuss, a gel analysis application currently under development (freely available at https://github.com/lkrippahl/eReuss), the migration distances for each concentration ratio were computed from the digital image of the electrophoresis gel by fitting Gaussian curves to the image intensity profiles averaged for each lane. This allowed a more reliable quantification of band migration, since the most relevant bands were very broad. This behavior was previously observed for BSA binding to AuNP, and data was fitted to a Hill-type adsorption isotherm (Equation (1)), using OriginPro9 software.
(1)$$\Delta \mu =\frac{\Delta \mu_{\max}\cdot {\left[\mathrm{anti}-\mathrm{HRP}\right]}^{n}}{K_{D}{}^{n}+{\left[\mathrm{anti}-\mathrm{HRP}\right]}^{n}}$$
in which Δµ is the variation of electrophoretic mobility between the data point and the AuNSs conjugate before addition of any antibody, and K_D_ is the dissociation constant (in M) corresponding to the value of the anti-HRP concentration for one-half of Δµ_max_. In the Hill model, a cooperativity parameter, n, accounts for positive (n > 1) or negative (n < 1) cooperativity, when the binding of the next antibody is favored or unfavored, respectively, by the binding of the previous one. When n = 1, no cooperativity is present, and a Langmuir-type adsorption isotherm can describe the system. 

### 2.10. Raman and SERS Measurements

Raman measurements were performed in a Renishaw inVia Qontor micro-Raman spectrometer equipped with an air-cooled charge-coupled device (CCD) detector and an He–Ne laser operating at 32 mW of 633 nm laser excitation. The spectral resolution of the spectroscopic system is 0.3 cm^−1^. The laser beam was focused with a 5× Leica objective lens (N Plan EPI) with a numerical aperture of 0.12. An integration time of 10 scans of 20 s each was used for all measurements to reduce the random background noise induced by the detector, without significantly increasing the acquisition time. The intensity of the incident laser was 3.2 mW. Triplicates were taken of all spectra. Between different Raman sessions, the spectrograph was calibrated using the Raman line at 521 cm^−1^ of an internal Si wafer for reducing possible fluctuations of the Raman system. A volume of 300 µL of each sample was deposited on a multi-well (n = 96) plate and the objective was focused inside the well. All SERS spectra were recorded at room temperature. All the raw data were collected digitally with Wire 5.0 software for processing. Vibrational line areas were determined with the aid of the Wire 5.0 software for all spectra.

## 3. Results and Discussion

The assembly scheme of the bioconjugates for the immunoassay is represented in Figure 1. The process consists of three consecutive steps: (1) synthesis of citrate capped gold nanostars, using spherical AuNPs as seeds, (2) functionalization with a Raman reporter (MBA) allowing SERS detection of the bioconjugates [11,39,40], (3) bioconjugation with anti-HRP antibodies for detection and BSA for blocking and increasing the specificity of antibody detection [27]. 

### 3.1. Characterization of the Synthesized Gold Nanoparticles

Gold nanoparticle samples were characterized by UV-Visible spectroscopy and SEM (Figure 2). Further UV-Visible, DLS, and XRD measurements allowed for determining diameters and colloidal concentrations (Appendix A). The UV-Vis spectrum after the synthesis of AuNPs shows a localized surface plasmon resonance (LSPR) band centered at approximately 519 nm. The low absorbance values at 600–700 nm indicate negligible colloid aggregation (Figure 2A). The morphology of the synthesized nanostars is characterized by a central core with multiple spikes, as confirmed by SEM (Figure 2B) [41]. The average tip-to-tip length observed was approximately 70 nm. AuNSs suspensions are typically polydisperse, with small differences in shape, length, orientation of spikes, and sharpness of their tips, as the SEM micrograph (Figure 2B) shows. Consequently, the optical response from AuNSs is complex, and originates from both the cores and the spikes. The high anisotropy gives rise to plasmonic modes of different orders, which produces hybridization of resonances at distinct wavelengths in the visible and near-IR [42,43]. The resonances observed around 720 and 800 nm can be assigned to dipolar plasmon modes and approximately 675 nm to multipolar plasmon modes [41]. The small broad resonance at approximately 530 nm is associated with transverse resonance within the branches or to plasmonic resonances of spherical protuberances. These spherical protuberances might be non-fully grown or reconstructed spikes and it is also possible that the spherical core contributes to the overall spectrum [32,44]. Thus, even within the same batch, the AuNSs have very heterogeneous morphologies, especially when compared with AuNPs, which justifies the UV-Vis spectrum with a broad peak shown in Figure 2A (blue line). The reproducibility among batches of AuNSs was studied on the same day of the synthesis and revealed good reproducibility (Appendix A). In Appendix A, stability of the AuNSs with time is presented, as analyzed by its UV-vis spectra (Appendix A) and SEM micrographs (Appendix A).

### 3.2. Functionalization with the Raman Reporter

The gold nanostars synthesized in this work are electrostatically stabilized in a colloid by adsorbed citrate anions [44]. Thiolated bifunctional linkers such as MBA, bind to the AuNSs surface by chemisorption through a terminal thiol group, which presents, at the other end, a carboxylic group that is deprotonated, and, therefore, negatively charged, at the pH of the conjugation reaction. Moreover, MBA is commonly used as a Raman reporter to allow the use of these bioconjugates in SERS-based immunoassays [39,45]. To determine the amount of MBA needed to form a monolayer around the NPs, AuNSs were functionalized with different MBA molar ratios (Figure 3). The functionalization process was assessed by UV-Vis spectroscopy by a 48-nm red-shift of the LSPR (Figure 3A), which is a consequence of an increase in the local refractive index at the AuNS surface [46]. The red-shift upon MBA functionalization for spherical AuNP is only 3 nm (data not shown). In fact, anisotropic NPs show a large change in the refractive index due to the presence of “hot spots” (especially at tips or edges) responsible for enhanced electromagnetic fields [22,24,46].

Confirmation of AuNSs functionalization with the Raman-active reporter was achieved by running SERS spectra of functionalized AuNSs for the different MBA to AuNSs molar ratios (Figure 3B). The Raman spectrum of MBA and the respective line assignments are presented in Appendix A. After extensive washing to remove non-bound MBA, SERS spectra of MBA-functionalized AuNSs, presented Raman spectra with two very intense vibrational lines at 1079 cm^−1^ and 1587 cm^−1^. The most intense lines also appeared in the MBA solution Raman spectrum (Appendix A) [47]. The SERS spectrum of the control citrate-capped AuNSs showed broad bands in the region of 1350–1450 cm^−1^, which are related to carboxylic group bending and stretching vibrations derived from the citrate molecules (Figure 3B) [48]. 

The electrophoretic mobility of NPs is affected by their mass, shape, and superficial charge [49]. Hence, this parameter can be used to verify the correct amount of the negatively charged MBA functionalization agent necessary to fully cover the NPs. Agarose gel electrophoresis is a particularly useful technique to study NPs and their bioconjugates since it uses simple equipment, and bands can be easily located without any dyeing step, due to the NPs’ intense colors (blue in the case of AuNS). In Figure 3C, an AGE for AuNSs functionalized with increasing molar ratios of MBA is shown. Two types of bands are observed in the AGE: (i) corresponding to aggregated AuNSs present in the gel wells, which did not migrate in the gel, and (ii) AuNSs presenting an electrophoretic mobility corresponding to a full MBA layer covering the AuNSs (Figure 3C). The lack of migration observed for non-functionalized AuNSs is likely caused by the loss of the weakly bound citrate ions or the presence of specific interactions between the gel and the NPs [50]. In order to guarantee that all the AuNSs were functionalized with a full MBA layer, a AuNS:MBA molar ratio of 1:50,000 was chosen, since, only in this case, all functionalized AuNSs migrated in the AGE, without any aggregation in the wells (see Figure 3C).

These results highlight the importance of confirming the formation of a Raman reporter monolayer at the AuNSs surface using complementary techniques.

### 3.3. Assembly of Antibody Conjugates

Raman reporter functionalized AuNS, were conjugated with anti-HRP. This polyclonal antibody was used to evaluate the parameters needed to perform conjugation, and as a model system for the proposed SERS-based immunoassays. Each tested molar ratio was performed in similar conditions maintaining the buffer and AuNS–MBA volume. In order to promote conjugation based mainly on establishing electrostatic interactions and considering that IgG immunoglobulins isoelectric point ranges from 6- to 9, using phosphate buffer at a pH of 7.2 is appropriate to guarantee that the antibody surface is mainly positively charged. These positive charges at the antibody surface, promote adsorption for the negatively charged nanoparticle surface provided by the carboxylic groups of MBA [50].

A 24-nm red-shift of the LSPR was observed upon bioconjugation with anti-HRP antibodies (see spectra in Appendix A). Additional proof of bioconjugation was obtained by AGE and used to assess AuNS–MBA–anti-HRP bioconjugates for various molar ratios of the antibody (Figure 4). The first well corresponds to AuNS–MBA, that, having a negative charge, migrate toward the positive pole. Upon anti-HRP binding, the hydrodynamic size increases and the global net charge can be partially cancelled. Consequently, the AuNS–MBA–anti-HRP bioconjugates migrate less toward the positive pole. Since the molar ratio at which anti-HRP is incubated with AuNS–MBA increases, a higher number of antibodies is expected to adsorb to the surface, which leads to a decrease in the electrophoretic mobility. The anti-HRP solution concentration corresponding to the stabilization of the electrophoretic mobility (plateau), is assumed to be the concentration necessary to form a full corona at the AuNS–MBA surface. The intermediate ratios possibly present smeared bands due to the variability in conjugation stages. Since the anti-HRP to AuNS–MBA conjugation process is a dynamic adsorption equilibrium, the concentration of the antibody at intermediate ratios is not sufficient to fully cover the AuNS–MBA, which gives rise to multiple equilibria. Moreover, the antibody molecule orientation upon adsorption might vary more at these intermediate ratios. This effect is more evident when the protein has more positively charged groups [51]. These wider bands can be more or less evident due to some variation during the experiment such as small variations in AuNSs concentration (loss during resuspension). Different concentrations of agarose gels are affected by different humidity levels and air exposures. Higher concentrations of agarose decrease the pore size, which leads to a stronger opposition to NPs movement. When the antibody concentration increases, bands become narrower, as a result of antibody saturation of the AuNS surface. It is important to stress that the conjugation ratio does not correspond to the total antibody at the AuNSs surface, but, rather, to the amount of antibody present in the incubation sample.

Furthermore, electrophoretic mobilities for several different AGE were calculated with the eReuss software and fitted to an adsorption isotherm of the Hill type (Figure 4B), with a dissociation constant, K_D_ = (38 ± 7) nM of anti-HRP, and n = 1.0 ± 0.1. The latter indicates that anti-HRP antibodies bind to AuNS–MBA in a non-cooperative manner, and basically follow a Langmuir adsorption model (see Materials and Methods for details on the fitting equation). The value obtained for the dissociation constant is of the same order of magnitude as the one found for binding another IgG antibody to 30 nm citrate-capped spherical AuNPs [12]. This is about two orders of magnitude lower than the dissociation constant for the binding of bovine serum albumin to 15 nm citrate-capped AuNP, which is a reference system for strong protein-AuNP binding [14]. Although these values cannot be directly compared since they refer to protein binding to gold NPs with different morphologies and concentrations, the low value obtained for the anti-HRP to AuNS–MBA dissociation constant, implies the formation of rather stable bioconjugates. Analyzing the conjugation fitting curve, and, in order to guarantee a full antibody corona at the AuNS surface, further assembly of anti-HRP containing conjugates was performed with the concentration of anti-HRP at 211 nM, for a constant AuNS–MBA concentration of 0.2 nM.

### 3.4. Blocking of Antibody Conjugates

In order to guarantee a specific and interference-free HRP antigen binding to the anti-HRP antibodies, a BSA blocking step was performed [51]. A concentration of BSA of 211 nM (same as anti-HRP antibody) was added, affording blocked bioconjugates with lowered electrophoretic mobility after washing excess BSA by centrifugation. Increase of the amount of BSA added to the anti-HRP loaded AuNS–MBA conjugates did not change their electrophoretic mobility further, so the concentration of BSA added in further experiments was always equal to the anti-HRP concentration. UV-Vis spectroscopy, SEM, DLS, and ζ-Potential measurements of the bioconjugates were also obtained to further characterize bioconjugates formation (Appendix A).

### 3.5. Competence of the Bioconjugates for SERS Immunoassays

The competence of the bioconjugates (AuNS–MBA–anti-HRP–BSA) for use in the SERS immunoassay was determined in two ways: (i) functionality of the adsorbed anti-HRP antibodies, i.e., their capacity to bind the respective antigens (the “analyte” in immunoassay), and (ii) SERS activity of the bioconjugates before and after binding their respective antigen, HRP.

#### 3.5.1. Functionality of the Adsorbed Anti-HRP Antibodies

The capacity of the AuNS–MBA adsorbed anti-HRP antibodies to bind to their respective antigens, was assessed by evaluating the peroxidase enzymatic activity, derived from bound HRP antigens, after incubating the bioconjugates with HRP and extensively washing non-bound antigens. Bioconjugates were prepared with four anti-HRP antibody concentrations, namely 53, 105, 211, and 422 nM, which correspond to two concentrations below and one above the one previously determined to guarantee a complete antibody corona in the bioconjugate (211 nM, see previous section). For each bioconjugate, a measured amount of HRP was added and, after 90 min of incubation time, washed by centrifugation. For comparison, the enzymatic activity of the same amount of free HRP was also determined. Results indicated that the prepared bioconjugates do contain functional anti-HRP antibodies, since increasing the amount of antibodies within the conjugate, leads to a higher amount of HRP capture, which translates into higher enzymatic activity for the bioconjugate (Figure 5). 

The detected activity increased from bioconjugates with 53 nM of anti-HRP to 105 nM and increased more to 211 nM, staying practically constant for bioconjugates prepared with an anti-HRP concentration of 422 nM. As such, an anti-HRP concentration of 211 nM was chosen to prepare future bioconjugates, since it represents a considerable saving in anti-HRP used to prepare active bioconjugates.

The remaining enzymatic activity was partially present in the respective supernatants (Appendix A). In fact, activities of supernatants that resulted from the centrifugation of the conjugates, revealed a slight increase for lower anti-HRP concentrations presented in the bioconjugates (Appendix A), with the two highest anti-HRP concentrations resulting in a lower activity in the supernatant, likely due to a higher concentration of HRP being captured by the bioconjugate. 

#### 3.5.2. SERS Activity of the Bioconjugates

Several dilutions from the stock suspension of bioconjugates alone, and bioconjugates incubated with HRP antigen and washed, were made and their SERS spectra were obtained. The detection was considered successful when both dominant vibrational lines for MBA were identified, at 1079 cm^−1^ and 1587 cm^−1^. The vibrational mode at 1079 cm^−1^, assigned to the C–S stretching, is expected to be strong, since the gold surface has high affinity to thiol groups provided by the MBA [47]. As expected, reducing the concentration of the bioconjugates lowers the observed intensity in the SERS spectra (Figure 6). The analytical area threshold for the simulation software (see Materials and Methods) to detect the 1079 cm^−1^ line of MBA was 0.5 a.u. This value corresponds to the spectrum of bioconjugates, with or without HRP, at 0.01 nM. These HRP-containing bioconjugates at 0.01 nM, present a peroxidase activity of 50 µU, which indicates a highly sensitive assay.

### 3.6. Proof-of-Concept for Using the Bioconjugates in the Immunoassay

Agarose gel electrophoresis was again crucial in establishing a proof-of-concept for the proposed SERS-based immunoassay in which we intend to use the assembled bioconjugates described. In that future assay, we intend to immobilize the same antibodies on a surface, such as a microfluidics platform. Figure 7 presents electrophoretic mobility as determined by AGE for the different stages of bioconjugation. The green bar corresponds to the sandwich complex, formed during antigen detection (positive result). This complex is bulkier and less negatively charged and so migrates less in the gel. The red bar corresponds to the negative result, in which anti-HRP antibodies fail to form the sandwich complex due to the absence of the antigen. As such, the same electrophoretic mobility is observed, which is similar to the bioconjugate alone (second grey bar). 

## 4. Conclusions

In this work, the seed-mediated growth of AuNSs allowed highly branched nanostructures, as confirmed by UV-Vis spectroscopy and SEM. A general approach was designed for a simple and effective functionalization of the AuNSs by a Raman reporter, and further conjugation with antibodies for final use as a SERS immunoassay. AGE was instrumental in determining the appropriate amounts of the Raman reporter MBA and antibody to AuNSs, in order to fully cover the AuNS surface. An MBA to AuNSs molar ratio of 50,000, showed excellent electrophoretic mobility without any signs of aggregation. The electrophoretic migration pattern of the conjugation samples (AuNS–MBA-anti-HRP) obtained by simple incubation, was fitted to the Langmuir model achieving a dissociation constant, K_D_ = (38 ± 7) nM of anti-HRP, which indicates the formation of stable bioconjugates, even though they were obtained by simple incubation. Furthermore, analysis of the Langmuir curve indicated that the conjugation condition provides a complete coverage of the AuNSs corresponding to an anti-HRP concentration of 211 nM added during incubation.

The functionality of the bioconjugates with anti-HRP is easy to determine since the respective antigen (HRP) presents peroxidase enzymatic activity. After bioconjugates incubation with the respective antigen, peroxidase activity determined after washing revealed that the bioconjugated anti-HRP antibodies, were competent for antigen binding. The optimized SERS-active bioconjugates, due to their anisotropic nanostructures, present several “hot spots” at tips and edges, which lead to an effective SERS detection down to 0.01 nM of bioconjugates. These present a peroxidase activity of 50 µU. All steps of conjugation were fully characterized by UV–Visible spectroscopy, DLS, ζ-Potential, SEM, and AGE. The latter technique was used to establish a proof-of-concept for the proposed immunoassay, including the sandwich binding of a second antibody to the HRP-loaded bioconjugates, which denotes a positive result for the immunoassay.

The fact that the antibody used, anti-HRP, represents a generic IgG antibody, opens various applications using the same detection principles but employing different antibody–antigen systems. Results presented in this paper are a proof-of-concept for a SERS-based immunoassay with easy adaptation to a microfluidics platform.

## Figures and Tables

**Figure 1 nanomaterials-09-01561-f001:**
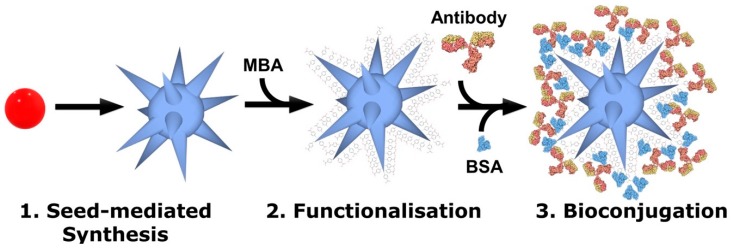
Schematic representation of the process to produce SERS-active bioconjugates. First, citrate-capped gold nanospheres are synthesized and used as seeds for gold nanostars synthesis. Second, the nanoparticles are functionalized with a Raman reporter. Third, bioconjugation with antibodies and blocking with bovine serum albumin (BSA) are completed by simple incubation.

**Figure 2 nanomaterials-09-01561-f002:**
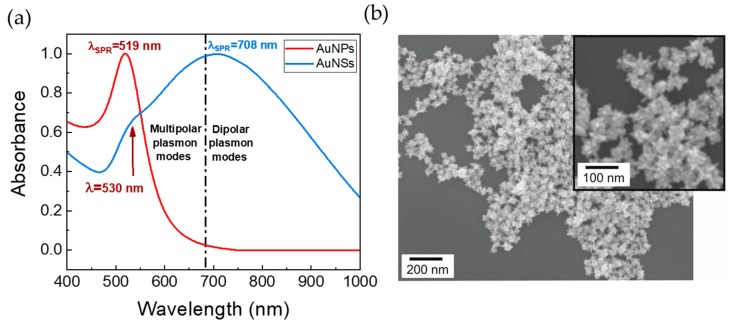
(**a**) UV-Vis spectra of AuNPs (red line) and AuNSs (blue line). The Ojea-Jiminéz [29] synthesis method allowed a diameter of 12 nm ($\lambda_{LSPR}=519\;\mathrm{nm}$) for AuNPs. The AuNSs present an LSPR band centered approximately at 670 nm and a large broadening. (**b**) SEM micrograph of AuNSs after synthesis.

**Figure 3 nanomaterials-09-01561-f003:**
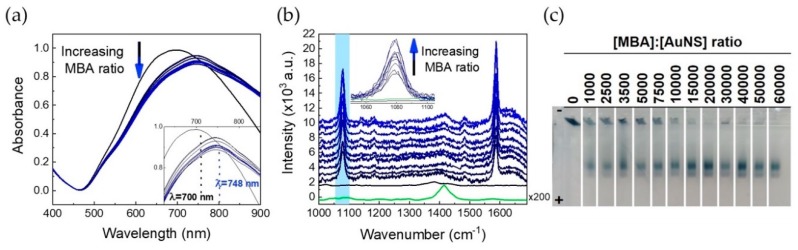
Analysis of MBA-functionalized AuNSs by UV-Vis spectroscopy, SERS, and AGE. (**a**) Normalized UV-Vis spectra of AuNSs functionalized with increasing MBA molar ratios. Comparing the LSPR bands of AuNSs capped with citrate from the synthesis (black line), with AuNSs functionalized with increasing molar ratios of MBA (blue lines), a bathochromic shift of 48 nm is observed. (**b**) SERS spectra of functionalized AuNSs, presented increasingly more intense lines at 1079 cm^−1^ (inset) and 1587 cm^−1^, which confirmed changing the capping agent by disappearing citrate lines and appearing MBA lines, since the molar ratio of MBA increases. Citrate SERS spectrum is presented in green, for comparison (**c**). Photograph of an agarose gel with AuNSs samples functionalized with increasing molar ratios of MBA.

**Figure 4 nanomaterials-09-01561-f004:**
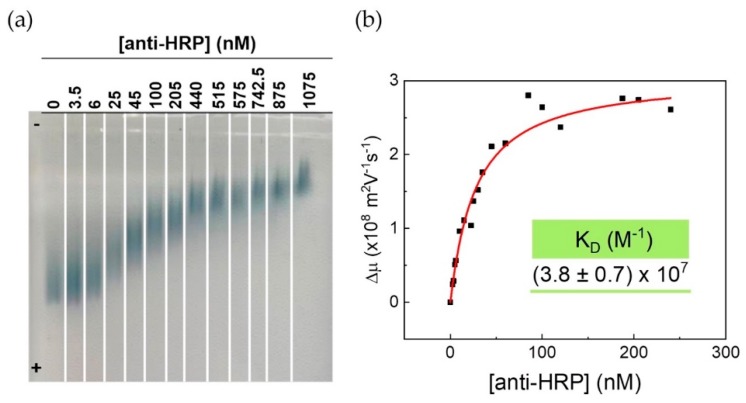
AGE of AuNS–MBA with increasing molar ratios of anti-HRP. (**a**) Exemplary 0.3% agarose gel for concentrations of anti-HRP from 3.5 to 1075 nM corresponding to [anti-HRP] to [AuNS–MBA] ratios from 7 to 2150. (**b**) Variation of electrophoretic mobility (Δµ) vs. anti-HRP concentration of the various AuNS–MBA-anti-HRP conjugates. Data was fitted to an adsorption isotherm of the Hill type with parameters Δµ_max_ = (3.1 ± 0.2) × 10^−8^ m^2^ V^−1^ s^−1^ K_D_ = (38 ± 7) nM, n = 1.0 ± 0.1, and adjusted R^2^ = 98%. Electrophoretic mobility is normalized for the maximum mobility, corresponding to the sample without the antibody. Data presented in the graph is derived from three different AGE, with each using a different set of anti-HRP concentration values.

**Figure 5 nanomaterials-09-01561-f005:**
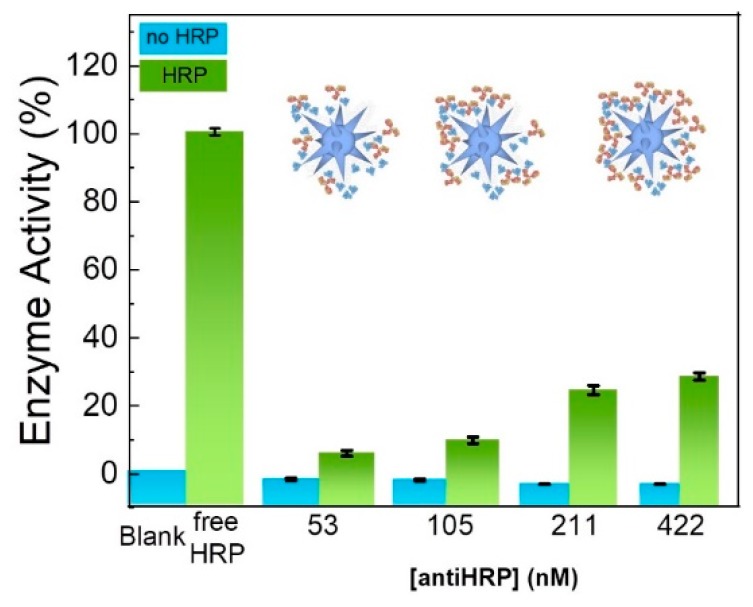
Enzymatic assays for bioconjugates. “Blank”: assay without HRP. “Free-HRP”: assay with the same amount of free HRP as added to the bioconjugates. The enzymatic activities for bioconjugates with and without HRP confirmed the positive recognition from the AuNS–MBA-immobilized anti-HRP antibody.

**Figure 6 nanomaterials-09-01561-f006:**
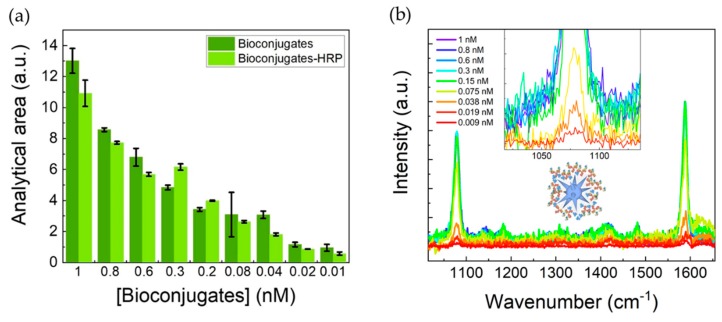
(**a**) SERS line areas for bioconjugates with and without a bound HRP antigen. Analytical areas presented are from the 1079 cm^−1^ line of MBA. (**b**) SERS spectra from all the performed dilutions for bioconjugates with bound HRP antigens. These concentrations are related to the concentration determined for AuNSs after synthesis.

**Figure 7 nanomaterials-09-01561-f007:**
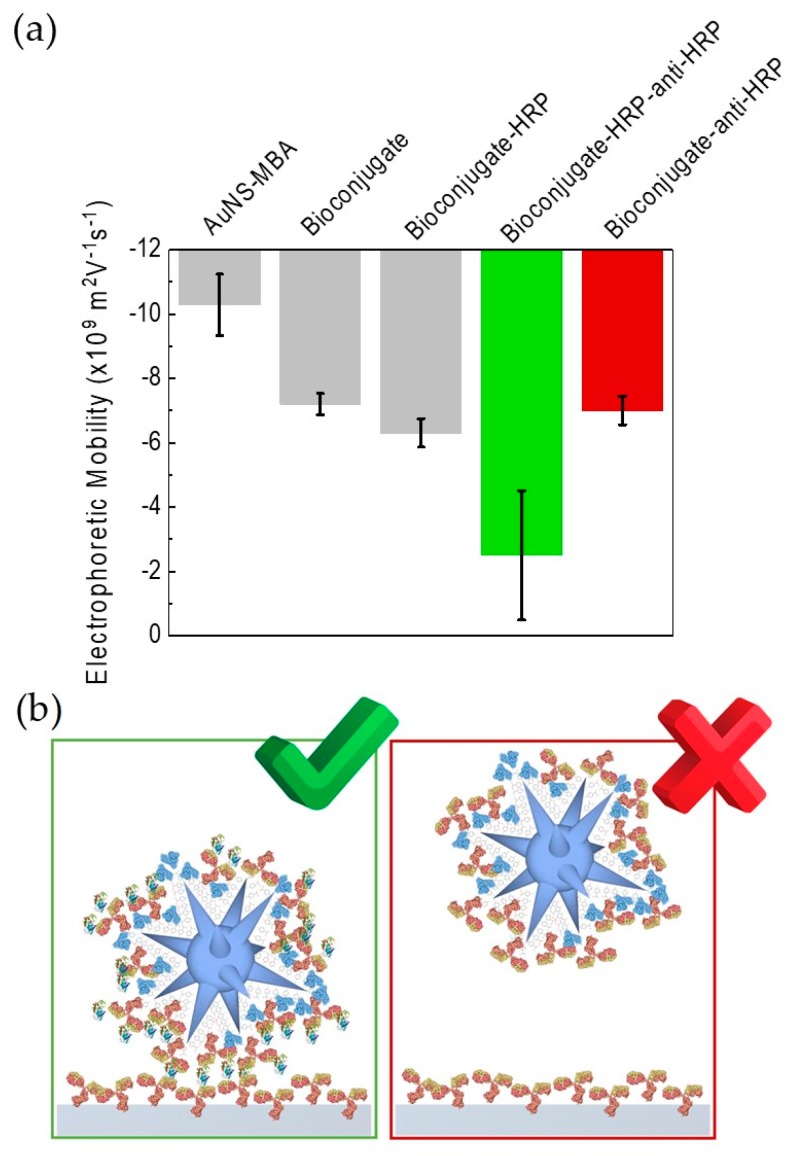
Proof-of-principle of the SERS-based immunoassay. (**a**) Electrophoretic mobility as determined by AGE for the different stages of bioconjugation. The green bar, corresponds to the sandwich complex, formed during HRP antigen detection (positive result). The red bar corresponds to the negative result, when no HRP antigen is present. Error bars correspond to triplicate experiments of three different sets of bioconjugates each. (**b**) Schematics of the SERS-based immunoassay at the molecular level.

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
