# Peer review of "Design and Simple Assembly of Gold Nanostar Bioconjugates for Surface-Enhanced Raman Spectroscopy Immunoassays"

_nanomaterials, 2019, doi:10.3390/nano9111561_

Round 1

Reviewer 1 Report

Thank you for the opportunity to review this interesting manuscript, which design and simple assembly of gold nanostar bioconjugates for Surface-Enhanced Raman Spectroscopy immunoassays

I consider this article to be well conducted, and well-written. 

The work is interesting and it represents a step forward in the scientific literature.

The topic is suitable for the publication on the Nanomaterials Journal.

The abstract is complete and describes briefly the main results and conclusions.

Considerations about the sections:

The introduction is well organized and reports the principal pieces of information. However, the aim of work is not clear. I suggest including a paragraph with the objective of the work at the end of the Introduction Section.

Author Response

The introduction is well organized and reports the principal pieces of information. However, the aim of work is not clear. I suggest including a paragraph with the objective of the work at the end of the Introduction Section.

We thank the reviewer for their observations. In order to clarify the objective of the work, the last paragraph of the introduction section has been changed:

“In this work, we demonstrate a simple method for optimising and testing bioconjugates for SERS-based immunoassays. Our bioconjugates consisted of AuNSs with a monolayer of the thiol and Raman reporter 4-mercaptobenzoic acid (MBA). In this way, MBA confers a distinct spectral signature and simultaneously a stabilising coating to the nanoparticles and a surface for conjugation with a molecular recognition element, the antibody. Electrostatic interactions were promoted by the carboxylic group of MBA and the amine group of antibody. The polyclonal antibody chosen was anti-Horseradish peroxidase (Anti-HRP) in order to easily access the viability of the bioconjugate through an enzymatic assay with horseradish peroxidase, which also functions as the antigen analyte [1]. Finally, the SERS signal of the bioconjugates was analysed to determine the minimum concentration detected. Agarose gel electrophoresis (AGE) was used to easily optimise the amount of antibody to load onto the AuNSs. Moreover, AGE allowed to establish a proof-of-concept for the proposed sandwich immunoassay. The bioconjugates formed in aqueous solution were characterised by ultraviolet-visible spectroscopy (UV-Vis), DLS and -potential measurements.”

Reviewer 2 Report

The work entitled: "Design and simple assembly of gold nanostar bioconjugates for Surface-Enhanced Raman Spectroscopy immunoassays" showed very important problem in detection the substances using Raman spectroscopy and potential of SERS. The construction of manuscript is good, however, I have several comments:

In the introduction Authors wrote: "The presence of several tips provides hot spots making them an ideal candidate for SERS applications". Authors should read the work Int. J. Mol. Sci. 2019, 20, 5011; doi:10.3390/ijms20205011, where it is shown the ability to control the lenght of the star arms. In the section reagents authors should describe all chemical compounds, which were used in the AuS NPs synthesis.  Why Authors used the 4-mercaptobenzoic acid?  What was the spped in the UV-Vis measurements? Authors should showed UV-Vis spectra od AuS NPs after biofunctionalization. SEM image should be changed, becaused in this figure the interaction between electron beam and NPs is visible. The conclusions should be shorter.

Reviewer 3 Report

The article by Oliveira et al. (Design and simple assembly of gold nanostar bioconjugates for Surface-Enhanced Raman Spectroscopy immunoassays) reports the fabrication of the immunoassay system using Surface-Enhanced Raman Spectroscopy. For this, they prepared bioconjugate system of polyclonal antibodies and anisotropic gold nanostars functionalized with a Raman reporter (4-mercaptobenzoic acid). They employed the Ultraviolet-Visible Spectroscopy, Dynamic Light Scattering, ζ-Potential, Scanning Electron Microscopy and Agarose Gel Electrophoresis to analyze the proposed system. I recommend the publication of the manuscript with some minor modifications. There is no labelling for the image on page 2. Is it an figure or something else? Please do not use the same abbrevations again and again through the manuscript. Use the abbrevation when it is first time used. Line 200-201, 205-207 These two statements seem to be same. Please merge these statements. Sample preparation for the Raman measurement is not mentioned in the experimental section. Please clearify this issue. There is no b in the Figure S9. Also, the color of the bars is the same. Please use different colors for the better evaluation. It would be better to add some SERS spectra of bioconjugate system in different concentratins (some spectra of Figure 6).
